# A Study on the Extraction, Fermentation Condition Optimization, and Antioxidant Activity Assessment of Polysaccharides Derived from *Kluyveromyces marxianus*

**DOI:** 10.3390/foods14162796

**Published:** 2025-08-12

**Authors:** Ziyin Xu, Lin Xu, Mei Chen, Zhonghai Li

**Affiliations:** State Key Laboratory of Green Papermaking and Resource Recycling, School of Bioengineering, Shandong Provincial Key Laboratory of Microbial Engineering, Qilu University of Technology, Shandong Academy of Sciences, Jinan 250353, China; xzy20011232022@163.com (Z.X.); 15762387136@163.com (L.X.); chenmei_16@163.com (M.C.)

**Keywords:** antioxidant capacity, exopolysaccharides, fermentation optimization, *Kluyveromyces marxianus*

## Abstract

*Kluyveromyces marxianus* exhibits advantages such as strong thermotolerance, rapid growth rate, and high safety, making it an excellent host cell for the production of bio-based products. In this study, two exopolysaccharides, KE1 and KE2, were isolated from the fermentation broth of the strain *K. marxianus* KM-502, and their hydroxyl radical scavenging, DPPH scavenging, and Fe^2+^-reducing activities were evaluated. In vitro antioxidant assays demonstrated that exopolysaccharide KE2 exhibited superior antioxidant activity compared to KE1. The fermentation conditions were optimized using single-factor experiments followed by response surface methodology (RSM). The optimized fermentation process revealed that the most suitable fermentation medium consisted of 8% sucrose, 1.99% peptone, and 0.13% CaCl_2_, while the optimal fermentation conditions were a medium volume of 74 mL in a 300 mL flask, pH 6.7, an inoculum size of 1.99%, a temperature of 30 °C, a shaking speed of 160 r/min, and a cultivation time of 96 h. After optimizing the fermentation conditions of *K. marxianus* KM-502, the exopolysaccharide (EPS) yield reached 5842.42 mg/L, representing a 22.77-fold increase compared to the yield before optimization. In summary, this study isolated exopolysaccharides KE1 and KE2 from *K. marxianus* KM-502. These exopolysaccharides demonstrated significant antioxidant activities, and the fermentation conditions for exopolysaccharide production were optimized. The findings of this study will facilitate the further development and utilization of exopolysaccharides from *K. marxianus*.

## 1. Introduction

Exopolysaccharides (EPS) are biopolymers secreted by microorganisms onto the cell surface, originating from diverse sources, including bacteria, fungi, and microalgae/cyanobacteria [1,2]. These compounds play an indispensable role in sustaining microbial viability under adverse conditions. For instance, EPS contribute to stabilizing cell membrane integrity, shielding microorganisms from extreme environmental stressors such as desiccation, elevated temperatures, and ultraviolet (UV) radiation. Additionally, EPS act as a protective barrier by inhibiting lysozyme activity, facilitating nutrient storage, and enhancing tolerance to toxic heavy metals [1,2,3]. In recent years, with the deepening of research on EPS, their multifaceted biological activities, including antioxidant, anti-inflammatory, and antitumor properties, have been increasingly recognized. Such attributes render EPS highly promising for applications across agriculture, food, cosmetics, and pharmaceutical industries, positioning them as a focal point in contemporary biotechnological research.

In the pharmaceutical sector, the antitumor activity of EPS represents a critical research frontier. Cancer remains one of the most pressing global health challenges, necessitating the continuous exploration of safe and effective therapeutic agents. EPS exerts antitumor effects through multiple mechanisms: (1) inhibiting tumor initiation, (2) inducing cancer cell apoptosis, and (3) enhancing immune system functionality [4,5]. Studies have demonstrated that BSPS-1 and BSPS-2, two EPS fractions isolated from *Bacillus subtilis* LZ13-4, induce apoptosis in HepG2 cells by upregulating apoptosis-related proteins and activating intracellular apoptotic signaling pathways, thereby inhibiting cancer cell proliferation [6]. Similarly, cell-bound EPS (cb-EPS) derived from *Lactobacillus acidophilus* 606 promote autophagic cell death through Beclin-1 and GRP78 induction, while indirectly enhancing this process via Bcl-2 and Bak modulation, exhibiting significant antitumor effects against HT-29 colon cancer cells [7]. Beyond medical applications, microbial EPS demonstrates excellent suspension and rheological properties coupled with remarkable biocompatibility, making it highly valuable in food industry applications [8]. For instance, EPS produced by *Lactobacillus plantarum* HM47 isolated from human breast milk has emerged as a promising food additive due to its unique structural features, effectively improving product texture, viscosity, and water-holding capacity [9]. The biological diversity and functional versatility of EPS stem from variations in microbial biosynthesis capabilities, polysaccharide chain architectures, and metabolic regulatory mechanisms. These factors collectively contribute to the broad spectrum of EPS applications across biomedical and industrial domains.

As an important source of biopolysaccharides, yeasts produce EPS that exhibit advantageous properties, including low production cost, non-toxicity, and environmental friendliness. Notably, EPS derived from this yeast demonstrates superior extractability from culture media compared to bacterial-derived counterparts, rendering large-scale EPS production via yeast fermentation a subject of substantial research interest [10,11,12]. Compared with traditional strains such as *Pichia pastoris* [13] and *Saccharomyces cerevisiae*, *Kluyveromyces marxianus*, a probiotic yeast, exhibits unique biological traits that confer substantial industrial significance [14]. This species demonstrates a broad substrate range, enabling the utilization of diverse carbon sources for growth and metabolism. Such metabolic versatility enhances adaptability across varied industrial production environments [15]. Notably, *K. marxianus* displays exceptional thermotolerance and rapid growth kinetics, achieving high cell densities within short timeframes—features that shorten production cycles and improve process efficiency [16]. Furthermore, as a yeast listed on the European Food Safety Authority’s (EFSA) Qualified Presumption of Safety (QPS) list and recognized as Generally Recognized as Safe (GRAS) by the U.S. Food and Drug Administration (FDA), *K. marxianus* is approved for direct addition to food products. These attributes collectively provide a robust foundation for its safe application in food fermentation [17].

The production of EPS by *K. marxianus* via submerged fermentation is influenced by multiple fermentation parameters, including medium composition, temperature, pH, and dissolved oxygen. Optimizing these conditions can enhance both the yield and quality of EPS, thereby facilitating its applications across various industries. This study systematically optimized the fermentation conditions for EPS production by *K. marxianus* KM-502 using a combination of single-factor experiments and Box–Behnken response surface methodology (RSM). Initially, single-factor experiments were conducted to identify the optimal types and concentrations of key fermentation parameters. Subsequently, Box–Behnken RSM was employed to investigate interactions among these factors and determine the optimal conditions. The optimized conditions significantly increased EPS yield compared to pre-optimization levels. Furthermore, two purified EPS fractions (KE1 and KE2) were isolated from the fermentation broth. In vitro antioxidant assays demonstrated that both KE1 and KE2 exhibited strong antioxidant activities. These findings provide experimental evidence supporting potential applications for promoting the application of *K. marxianus* EPS in cosmetic and food industries, thereby advancing the development and industrialization of EPS–based products.

## 2. Materials and Methods

### 2.1. Reagents and Chemicals

Yeast extract and peptone were purchased from Beijing Aoboxing Bio-tech Co., Ltd. (Beijing, China). DEAE-52 Cellulose and Sephadex G-100 were obtained from Beijing Sunny Biotechnology Co., Ltd. (Beijing, China). 2,2-Diphenyl-1-picrylhydrazyl (DPPH) was sourced from Macklin Biochemical Co., Ltd. (Shanghai, China). 2,4,6-Tris(2-pyridyl)-s-triazine (TPTZ) was acquired from Yuanye Bio-Technology Co., Ltd. (Shanghai, China).

### 2.2. Strains and Media

#### 2.2.1. Strains

*Kluyveromyces marxianus* KM-502, hereafter referred to as KM-502, represents a specific strain of *K. marxianus* isolated from *Tibetan Kefir grains* by our research laboratory.

#### 2.2.2. Cultivation Conditions for Strain

The KM-502 strain, which had been stored at −80 °C, was streaked onto a YPD solid plate for activation. Subsequently, a single colony was picked and inoculated into YPD liquid medium. The culture was then incubated at 30 °C with shaking at 200 rpm for 12 h. Afterward, the culture was transferred to fresh YPD liquid medium under the same incubation conditions and cultured until it reached the logarithmic (exponential) growth phase. Finally, the culture was transferred to fermentation medium and incubated at 30 °C with shaking at 160 rpm for 96 h to initiate fermentation.

#### 2.2.3. Culture Media

YPD medium: This medium was composed of 1% yeast extract, 2% glucose, and 2% peptone. It was sterilized at 115 °C for 30 min.

YPD solid medium: This medium was composed of 1% yeast extract, 2% glucose, 2% peptone, and 1.5% agar powder. It was also sterilized at 115 °C for 30 min.

### 2.3. Fermentation Optimization

#### 2.3.1. Single-Factor Screening of Fermentation Medium Components

To identify the optimal carbon source, nitrogen source, and inorganic salt ions for the fermentation medium, our experiments were based on the YPD liquid medium. We selected 2% glucose, xylose, lactose, fructose, and sucrose as carbon sources; 1% yeast extract, peptone, a mixture of yeast extract and peptone (1:1), a mixture of yeast extract and ammonium sulfate (1:1), and a mixture of peptone and ammonium sulfate (1:1) as nitrogen sources; and 0.05% NaCl, MgSO_4_, and CaCl_2_ as inorganic salts. The medium volume was set at 40% of the flask capacity, the inoculum size was 4%, the natural pH of the fermentation broth was maintained, the culture temperature was 30 °C, the shaking speed was 160 r/min, and the fermentation time was 96 h. The concentration of exopolysaccharides was determined using the phenol–sulfuric acid method. Each experiment was conducted with three biological replicates.

#### 2.3.2. Single-Factor Concentration Screening of Fermentation Medium Components

To determine the optimal concentrations of sucrose, peptone, and CaCl_2_ in the fermentation medium, we again based our experiments on the YPD liquid medium. The concentrations of the optimal carbon source (sucrose) were set at 1%, 2%, 4%, 6%, and 8%; the concentrations of the optimal nitrogen source (peptone) were set at 0.25%, 0.5%, 1%, 2%, and 3%; and the concentrations of the optimal inorganic salt ion (CaCl_2_) were set at 0.05%, 0.1%, 0.2%, 0.4%, and 0.6%. The medium volume was set at 40% of the flask capacity, the inoculum size was 4%, the natural pH of the fermentation broth was maintained, and the culture was carried out in a shaker at 30 °C and 160 r/min for 96 h. The concentration of exopolysaccharides was determined using the phenol–sulfuric acid method. Each experiment was conducted with three biological replicates.

#### 2.3.3. Box–Behnken Design for Screening the Optimal Fermentation Medium

The optimal carbon source, nitrogen source, and inorganic salt ions, labeled A, B, and C, respectively, were selected as significant factors and independent variables. Based on the optimal concentrations of individual components, a three-level coded design was conducted for each independent variable. The concentration of extracellular polysaccharides in the fermentation broth was used as the response value for the response surface methodology (RSM) experimental design. The Design Expert software (v13.0.5.0) was employed to select a central composite design with N = 17 experimental runs.

#### 2.3.4. Single-Factor Screening of Fermentation Conditions

To screen for the optimal initial pH of the fermentation broth, inoculum size, and medium volume for exopolysaccharide production, the optimal medium identified through the Box–Behnken central composite design experiment was used as the basal fermentation medium. The initial pH of the fermentation medium was set at 4, 5, 6, 7, 8, and 9, respectively. The inoculum sizes were set at 0.5%, 1%, 2%, 3%, and 10%, respectively. The medium volumes in the flasks were set at 10%, 20%, 30%, 40%, and 50%, respectively. The fermentation was carried out in a shaking incubator at 30 °C and 160 r/min for 4 days. The concentration of extracellular polysaccharides was determined using the phenol–sulfuric acid method. Each experiment was performed with three biological replicates.

#### 2.3.5. Box–Behnken Design for Screening the Optimal Fermentation Conditions

The optimal medium volume, initial pH, and inoculum size of the fermentation broth were selected as significant factors and independent variables, labeled A, B, and C, respectively. Based on the optimal values of individual components, a three-level coded design was conducted for each independent variable. The concentration of extracellular polysaccharides in the fermentation broth was used as the response value for the response surface methodology (RSM) experimental design. The Design Expert software (v13.0.5.0) was employed to select a central composite design with N = 17 experimental runs.

### 2.4. Extraction and Purification of Polysaccharides

#### 2.4.1. Extraction of Polysaccharides

Extraction of polysaccharides from KM-502 fermentation broth: The fermentation supernatant was concentrated to one-tenth of its original volume using a rotary evaporator. Subsequently, 95% ethanol was added to the concentrated solution in a 1:3 ratio (concentrated solution volume:95% ethanol volume). The mixture was thoroughly mixed and allowed to stand at 4 °C for 12 h. Following centrifugation, the precipitate was collected and subjected to vacuum freeze-drying to obtain crude polysaccharides. The dried crude polysaccharides were then dissolved in water, and proteins in the polysaccharide solution were removed using the Sevag method [6]. The supernatant was dialyzed using a dialysis membrane with a molecular weight cut-off of 1000 Da. The dialyzed polysaccharide solution was vacuum freeze-dried to yield purified polysaccharides, which were stored at −20 °C.

#### 2.4.2. DEAE-52 Cellulose Column

The extracted purified polysaccharides were weighed and dissolved in ultrapure water to prepare a 20 mg/mL polysaccharide solution. After centrifugation to remove any precipitate, the supernatant was filtered through a 0.22-micron organic filter membrane and loaded onto a DEAE-52 cellulose column (2.6 cm × 20 cm) for purification. Polysaccharides were eluted from the column using a NaCl concentration gradient. The first 20 collection tubes were eluted with distilled water, whereas tubes 21–80 were eluted with a linear gradient of 0–1 mol/L NaCl solution. Each tube collected 4.5 mL of eluate. The polysaccharide concentration in each tube was detected at 495 nm using the phenol–sulfuric acid method. The polysaccharide solutions corresponding to the collection tubes exhibiting a single absorption peak were combined. The combined polysaccharide solution was then dialyzed using a dialysis membrane with a molecular weight cut-off of 1000 Da to remove NaCl. The dialyzed polysaccharide solution was freeze-dried.

#### 2.4.3. Sephadex G-100 Column

The polysaccharide sample obtained after DEAE-52 cellulose column chromatography was weighed and dissolved in ultrapure water to prepare a 20 mg/mL polysaccharide solution. After centrifugation to remove any precipitate, the supernatant was filtered through a 0.22 μm organic filter membrane and loaded onto a Sephadex G-100 column (1.6 cm × 70 cm) for purification. Polysaccharides were eluted from the column with ultrapure water. Each tube collected 3 mL of eluate. The polysaccharide concentration in each tube was detected at 495 nm using the phenol–sulfuric acid method. The polysaccharide solutions corresponding to the collection tubes showing a single absorption peak were combined. Finally, the combined polysaccharide solution was freeze-dried.

### 2.5. In Vitro Antioxidant Activity of Polysaccharides

#### 2.5.1. Hydroxyl Radical Scavenging Activity

The method for measuring the hydroxyl radical scavenging activity of polysaccharide samples described in Ref. [18] was used, with slight modifications. The polysaccharide solutions were prepared at concentrations of 2 mg/mL, 4 mg/mL, 6 mg/mL, 8 mg/mL, and 10 mg/mL, respectively. Then, 1 mL of polysaccharide solution at each concentration was taken and mixed with 1 mL of 9.0 mmol/L FeSO_4_ solution, 1 mL of 9.0 mmol/L salicylic acid–ethanol solution, and 1 mL of 8.8 mmol/L H_2_O_2_ solution. The reaction mixture was incubated in a 37 °C water bath for 30 min and then centrifuged to remove any precipitate. In the blank group, ultrapure water was used instead of the *K. marxianus* polysaccharide solution. In the control group, distilled water was used instead of the 8.8 mmol/L H_2_O_2_ solution. Each group was subjected to three replicate experiments. The absorbance values of the polysaccharide reaction solutions were measured at a wavelength of 510 nm. Formula: hydroxyl radical scavenging activity (%) = [*A_x_* − (*A*_1_ − *A*_2_)]/*A_x_* × 100, where *A*_1_ was the absorbance of the sample group (polysaccharide solutions at different concentrations), *A*_2_ was the absorbance of the control group (distilled water substituted for 8.8 mmol/L H_2_O_2_, and *A_x_* was the absorbance of the blank group (distilled water substituted for polysaccharide solution).

#### 2.5.2. DPPH Radical Scavenging Activity

The method for assessing the DPPH radical scavenging capacity of polysaccharide samples described in Ref. [19] was used, with slight modifications. Polysaccharide solutions were prepared at concentrations of 2 mg/mL, 4 mg/mL, 6 mg/mL, 8 mg/mL, and 10 mg/mL, respectively. Subsequently, 1 mL of each polysaccharide solution was mixed with 2 mL of a 0.1 g/L DPPH (1,1-diphenyl-2-picrylhydrazyl) ethanol solution. The reaction mixture was incubated at 25 °C for 60 min. In the blank group, ultrapure water was substituted for the *K. marxianus* polysaccharide solution. In the control group, absolute ethanol was used in place of the 0.1 g/L DPPH radical ethanol solution. The absorbance values of the polysaccharide reaction solutions were measured at a wavelength of 517 nm. Formula: DPPH radical scavenging activity (%) = [1 − (*A_i_* − *A_j_*)]/*A*_0_ × 100, where *A_i_* was absorbance of the sample group (polysaccharide solutions at different concentrations), *A_j_* was absorbance of the control group (distilled water substituted for 0.1 g/L DPPH), *A*_0_ was absorbance of the blank group (distilled water substituted for polysaccharide solution).

#### 2.5.3. Fe^2+^ Reduction Activity

The method used for assessing the extracellular ferric ion-reducing ability of polysaccharide samples was adapted from the approach described by Chang, with minor modifications [20]. The FRAP (Ferric-Reducing Antioxidant Power) reaction system comprises acetate buffer (300 mmol/L, pH 3.6), TPTZ solution (10 mmol/L), and FeCl_3_•6H_2_O solution (20 mmol/L). Polysaccharide samples were prepared at concentrations of 2 mg/mL, 4 mg/mL, 6 mg/mL, 8 mg/mL, and 10 mg/mL. Subsequently, 1 mL of each polysaccharide solution was mixed with 2.5 mL of the FRAP reagent. The reaction mixture was incubated at 37 °C for 8 min, after which the absorbance values of the polysaccharide reaction solutions were measured at a wavelength of 593 nm. The ferric ion-reducing ability of extracellular polysaccharides derived from *K. marxianus* was determined based on the standard curve of FeSO_4_•6H_2_O (0–100 μmol/L).

## 3. Results

### 3.1. Optimization of Fermentation Medium Components

#### 3.1.1. Single-Factor Screening of Fermentation Medium Components

Carbon sources, nitrogen sources, and inorganic salt ions are critical factors influencing the microbial fermentation synthesis of extracellular polysaccharides [21]. Among these, the type and concentration of the carbon source in the medium are closely related to the yield and physicochemical properties of extracellular polysaccharides. In this study, five carbon sources were evaluated: glucose, xylose, lactose, fructose, and sucrose. As illustrated in Figure 1a, when glucose was used as the carbon source, the extracellular polysaccharide yield of KM-502 was the lowest, at 255.1 mg/L. When xylose, lactose, and fructose were employed as carbon sources, the extracellular polysaccharide yields were relatively comparable, at 347.31 mg/L, 352.28 mg/L, and 338.83 mg/L, respectively. When sucrose was used as the carbon source, the extracellular polysaccharide yield of KM-502 reached its maximum, at 567.77 mg/L. Therefore, sucrose was identified as the optimal carbon source for the liquid fermentation of KM-502 to produce extracellular polysaccharides.

Nitrogen sources available to microorganisms can be categorized into organic nitrogen sources and inorganic nitrogen sources. In this study, various combinations of nitrogen sources were tested—including organic nitrogen sources alone and combinations of organic and inorganic nitrogen sources:yeast extract, peptone, yeast extract:ammonium sulfate (1:1); yeast extract:peptone (1:1); and peptone:ammonium sulfate (1:1)—to analyze their effects on the extracellular polysaccharide yield of KM-502. As depicted in Figure 1b, when peptone was used as the nitrogen source, the extracellular polysaccharide yield of KM-502 was the highest, at 570.31 mg/L. When peptone:ammonium sulfate (1:1) was used as the nitrogen source, the extracellular polysaccharide yield of KM-502 was the lowest, at 81.59 mg/L. The extracellular polysaccharide yields when yeast extract, yeast extract:ammonium sulfate (1:1), and yeast extract:peptone (1:1) were used as nitrogen sources were 322.5 mg/L, 255.86 mg/L, and 185.89 mg/L, respectively. Therefore, peptone was identified as the optimal nitrogen source for the liquid fermentation of KM-502 to produce extracellular polysaccharides, whereas peptone:ammonium sulfate (1:1) was not conducive to extracellular polysaccharide production by KM-502.

As illustrated in Figure 1c, under the same addition amount, when CaCl_2_ was used as the inorganic salt ion in the medium, the extracellular polysaccharide yield of KM-502 was the highest, reaching 577.34 mg/L, compared to 379 mg/L and 499.8 mg/L when NaCl and MgSO_4_ were used as inorganic salt ions, respectively. This suggests that calcium ions play a significant role in the synthesis and secretion of extracellular polysaccharides by KM-502.

#### 3.1.2. Effects of Sucrose, Peptone, and CaCl_2_ Concentrations on the Extracellular Polysaccharide Yield of KM-502

The effect of sucrose on the extracellular EPS yield of KM-502 is depicted in Figure 2a. The EPS yields exhibited significant variation across different sucrose concentrations. When the sucrose concentration was 6%, the EPS yield reached its maximum of 1190.33 mg/L. Further increasing the sucrose concentration in the medium resulted in a notable decline in the EPS yield. At a sucrose concentration of 8%, the EPS yield was 530.44 mg/L. It is inferred that the osmotic pressure generated by high sucrose concentrations may hinder EPS production during the growth and metabolism of KM-502. Therefore, the optimal sucrose concentration in the medium was determined to be 6%.

As illustrated in Figure 2b, within the peptone concentration range of 0.25% to 2%, the EPS yield of KM-502 increased with an increase in peptone concentration. When the peptone concentration was 2%, the EPS yield of KM-502 reached its highest value of 607.56 mg/L. However, as the peptone concentration continued to rise, the EPS yield decreased. At a peptone concentration of 3%, the EPS yield of KM-502 was 250.43 mg/L. This decrease may be attributed to the accumulation of nitrogen in the fermentation medium at higher peptone concentrations, which subsequently inhibits polysaccharide synthesis. Therefore, the optimal peptone concentration in the medium was selected as 2%.

According to Figure 2c, when the CaCl_2_ concentration was 0.1%, the EPS yield of KM-502 was 424.5 mg/L. Further increasing the CaCl_2_ concentration led to a decrease in the EPS yield. At a CaCl_2_ concentration of 0.2%, the EPS yield was 354.59 mg/L. When the CaCl_2_ concentrations were 0.4% and 0.6%, the EPS yields did not differ significantly, measuring 257.84 mg/L and 264.45 mg/L, respectively. Therefore, the optimal CaCl_2_ concentration in the medium was determined to be 0.1%.

#### 3.1.3. Optimization of Fermentation Medium Using the Box–Behnken Design

Based on the 17 sets of experimental conditions designed using the Box–Behnken design, the extracellular EPS yield of KM-502 was analyzed. According to the results presented in Table 1, the EPS yield ranged from 328.88 mg/L to 2833.88 mg/L, indicating that the composition of the fermentation medium significantly influenced the EPS yield of KM-502. Based on the variance data of the regression model in Table 2, the calculated multiple correlation coefficient (R^2^) was 98.77%, suggesting a high degree of fit for the polynomial model and a strong correlation between the predicted and actual EPS yield data. The F-value of the model was 62.39, with a corresponding *p*-value of 0.0001, indicating that the regression model was statistically significant. For the lack-of-fit term, the F-value was 4.74, with a *p*-value of 0.0835, which is higher than the 0.05 threshold, suggesting that the lack-of-fit effect was not significant. Therefore, this model can be used to analyze and predict the EPS yield of KM-502.

After performing regression analysis on the experimental data, a quadratic multiple regression equation was obtained: EPS = 2700.47 + 466.71A + 45.13B − 307.29C + 654.66AB − 22.31AC + 47.62BC − 210.87A^2^ − 1260.34B^2^ − 555.19C^2^, where A represents sucrose, B represents peptone, and C represents CaCl_2_. Using the response surface regression equation, response surface contour plots were generated using Design Expert 8.05 software. As illustrated in Figure 3, by comparing the pairwise interactions of sucrose (A), peptone (B), and CaCl_2_ (C) on EPS yield, it was observed that the interaction effect of BC was greater than that of AC, which in turn was greater than that of AB (BC > AC > AB).

According to the predictions from the response surface analysis, the optimal fermentation conditions for EPS production by KM-502 were determined to be 8% sucrose, 1.99% peptone, and 0.13% CaCl_2_. An experiment was designed and conducted under these conditions, revealing an EPS yield of 2942.53 mg/L, which was 9.12 times higher than the EPS yield obtained with the original fermentation medium. This result was consistent with the EPS yield predicted by the response surface experiment.

### 3.2. Optimization of Fermentation Conditions

#### 3.2.1. Single-Factor Screening of Fermentation Conditions for EPS Production by KM-502

Changes in the initial pH of the fermentation broth can cause variations in the solubility, molecular weight, and charge of the desired extracellular EPS [22,23]. As shown in Figure 4a, within the initial pH range of 4 to 7, the EPS yield of KM-502 increased with the rise in pH. The highest EPS yield of 3670.53 mg/L was achieved when the initial pH of the fermentation broth was 7. Subsequently, as the initial pH increased beyond 7, the EPS yield decreased, reaching a minimum of 767.96 mg/L at pH 9. It is speculated that excessively high pH values may affect the activity of key enzymes involved in the metabolic process of KM-502 and the permeability of the cell membrane, thereby reducing the EPS yield. These experimental results indicate that the initial pH of the fermentation broth significantly impacts the EPS yield of KM-502. Therefore, the optimal initial pH for EPS production by KM-502 is 7.

The effect of inoculum size on the EPS yield of KM-502 is illustrated in Figure 4b. When the inoculum sizes were 0.5%, 1%, and 2%, the EPS yield increased with the rise in inoculum size, reaching yields of 1915.25 mg/L, 2745.3 mg/L, and 3823.49 mg/L, respectively. However, when the inoculum size was increased to 3%, the EPS yield decreased to 3023.47 mg/L. Further increasing the inoculum size to 10% resulted in the lowest EPS yield of 1621.17 mg/L. It is hypothesized that an excessively high inoculum size accelerates the consumption of carbohydrate nutrients, which is unfavorable for EPS accumulation. Therefore, the optimal inoculum size for EPS production by KM-502 is 2%.

The volume of the fermentation broth in the flask affects oxygen transfer efficiency. As depicted in Figure 4c, when the volume of the fermentation broth was 10%, 20%, and 30% of the flask’s capacity, the EPS yield of KM-502 increased with the rise in volume, reaching yields of 1584.12 mg/L, 1721.37 mg/L, and 3100.37 mg/L, respectively. However, when the volume was further increased to 40% and 50%, the EPS yield decreased significantly to 2748.33 mg/L and 501.14 mg/L, respectively, compared to the yield at 30% volume. It is speculated that at low volumes, the large gas–liquid interface in the shake flask facilitates high oxygen diffusion rates, promoting aerobic metabolism in the yeast and providing sufficient ATP and reducing power for EPS synthesis. In contrast, high volumes restrict dissolved oxygen, forcing the yeast cells to switch to anaerobic fermentation, generating by-products such as ethanol and reducing the availability of precursors and energy required for polysaccharide synthesis. Therefore, the optimal volume of the fermentation broth for EPS production by KM-502 in liquid fermentation is 30% of the flask’s capacity.

#### 3.2.2. Optimization of Fermentation Conditions for EPS Production by KM-502 Using the Box–Behnken Design

Based on 17 sets of experimental conditions designed using the Box–Behnken design, we analyzed the extracellular EPS yields. The results presented in Table 3 indicate that the EPS yields ranged from 1482.35 mg/L to 5980.57 mg/L, suggesting that the composition of the fermentation medium significantly influences the EPS yield of *K. marxianus* strain KM-502.

According to the regression model variance data in Table 4, the calculated multiple correlation coefficient (R^2^) is 97.47%, indicating a high degree of fit for the polynomial model and a strong correlation between the predicted and actual observed data. The model F-value is 29.99, with a corresponding *p*-value of 0.0001, demonstrating the statistical significance of the regression model. For the lack-of-fit term, the F-value is 1.94, with a *p*-value of 0.2652, which is above the 0.05 threshold, indicating that the lack-of-fit effect is not significant. Therefore, this model can be reliably used to analyze and predict the EPS yield of strain KM-502.

Based on the experimental data, a quadratic multiple regression equation was obtained through regression analysis: EPS = 5586.25 − 801.23A − 77.94B − 184.18C + 742.92AB − 499.12AC + 363.08BC − 983.85A^2^ − 747.91B^2^ − 149.14C^2^, where A represents the volume of the medium in the flask, B represents the initial pH of the fermentation broth, and C represents the inoculum size of strain KM-502. Using the response surface regression equation, response surface contour plots were generated using Design Expert 8.05 software. By comparing the pairwise interactions of the three factors on EPS yield, as shown in Figure 5, it was found that the interaction is as follows: BC > AC > AB. According to the response surface predictions, the optimal conditions for EPS production by strain KM-502 are a medium volume of 74 mL in a 300 mL flask, an initial pH of 6.7, and an inoculum size of 1.99% of the fermentation broth volume. Under these conditions, the EPS yield reached 5842.42 mg/L, which is 22.77 times higher than the EPS yield obtained with the original fermentation medium and is consistent with the EPS yield predicted by the response surface experiment.

### 3.3. Purification of Polysaccharides

The crude polysaccharides extracted from the fermentation broth of KM-502 were purified using DEAE-52 cellulose column chromatography. The results are depicted in Figure 6a. The polysaccharide fractions corresponding to the single absorption peak were combined and freeze-dried, and the resulting polysaccharide was named KE0. Subsequently, polysaccharide KE0 underwent further purification using Sephadex G-100 column chromatography. The results are illustrated in Figure 6b. The eluates corresponding to the first and second absorption peaks detected during the polysaccharide analysis were collected and combined. These combined eluates were then freeze-dried separately to yield two purified polysaccharides, designated KE1 and KE2.

### 3.4. Extracellular Antioxidant Analysis of Polysaccharides

#### 3.4.1. Analysis of Hydroxyl Radical Scavenging Activity of Extracellular Polysaccharides

The hydroxyl radical scavenging abilities of extracellular polysaccharides KE1 and KE2 derived from KM-502 were assessed using the salicylic acid method, with ascorbic acid serving as the positive control. The experimental results, presented in Figure 7a, reveal that the hydroxyl radical scavenging activities of both KE1 and KE2 increased with increasing polysaccharide concentration within the range of 2 mg/mL to 10 mg/mL. At polysaccharide concentrations of 2, 4, 6, 8, and 10 mg/mL, the hydroxyl radical scavenging activity of KE1 was 5.67%, 13.4%, 21.12%, 22.05%, and 26.6%; for KE2, the capacities were 8.91%, 25.54%, 30.59%, 41.68%, and 48.71%, respectively. These findings suggest that extracellular polysaccharides KE1 and KE2 from KM-502 exhibit good hydroxyl radical scavenging abilities, with KE2 demonstrating significantly higher activity than KE1.

#### 3.4.2. Analysis of DPPH Scavenging Activity of Extracellular Polysaccharides

DPPH scavenging activity is a crucial parameter for evaluating the antioxidant performance of active substances. The extracellular DPPH scavenging activities of polysaccharides KE1 and KE2 from KM-502 are illustrated in Figure 7b. Within the polysaccharide concentration range of 2 mg/mL to 10 mg/mL, the DPPH scavenging activities of both KE1 and KE2 increased with increasing concentration. At polysaccharide concentrations of 2, 4, 6, 8, and 10 mg/mL, the DPPH· scavenging capacities of KE1 were 42.58%, 50%, 51.74%, 63.01%, and 63.71%; for KE2, the capacities were 43.87%, 55%, 74.19%, 74.36%, and 83.23%, respectively. These results indicate that extracellular polysaccharides KE1 and KE2 from KM-502 possess good DPPH scavenging abilities, with KE2 exhibiting significantly higher activity than KE1.

#### 3.4.3. Analysis of Fe^2+^-Reducing Activity of Extracellular Polysaccharides

Fe^2+^-reducing capacity is a widely used method for assessing the antioxidant activity of polysaccharides. Based on the standard curve for Fe^2+^-reducing capacity shown in Figure 7c, the regression equation obtained was y = 0.0003x + 0.00541, with an R^2^ value of 0.9992. Using ascorbic acid as the positive control, the extracellular Fe^2+^-reducing capacities of polysaccharides KE1 and KE2 were analyzed according to the aforementioned equation. The results, depicted in Figure 7d, indicate that the Fe^2+^-reducing activities of both KE1 and KE2 were concentration-dependent within the polysaccharide concentration range of 2 mg/mL to 10 mg/mL, increasing with increasing concentration. At polysaccharide concentrations of 2, 4, 6, 8, and 10 mg/mL, the Fe^2+^-reducing capacities of KE1 were 61.33 μmol/L, 118 μmol/L, 189.67 μmol/L, 243 μmol/L, and 321.33 μmol/L; for KE2, the capacities were 204.67 μmol/L, 413 μmol/L, 593 μmol/L, 833 μmol/L, and 1113 μmol/L, respectively. These findings suggest that extracellular polysaccharides KE1 and KE2 from KM-502 possess good Fe^2+^-reducing activities, with KE2 exhibiting significantly higher activity than KE1.

## 4. Discussion

The production of extracellular EPS through microbial fermentation is influenced by a multitude of factors, leading to variations in both the physicochemical properties and yield of EPS from the same strain under different fermentation conditions. Key factors such as carbon source, nitrogen source, inoculum size, and temperature significantly impact the production efficiency and economic viability of EPS [24]. Optimizing the fermentation system represents a crucial strategy for enhancing the yield of microbial EPS.

In this study, we conducted a comprehensive analysis of the effects of various carbon and nitrogen sources, as well as exogenous growth factors, on EPS yield. This analysis enabled us to derive an optimal formulation for the liquid fermentation medium. Subsequently, we employed the Box–Behnken design for response surface methodology (RSM) optimization experiments to elucidate the interactions among these factors. Based on the optimal medium predicted by RSM, we further investigated the effects of pH, medium volume, and inoculum size on EPS production by KM-502, thereby determining the optimal fermentation conditions. Another Box–Behnken design RSM optimization experiment was then conducted to analyze the interactions among factors, ultimately identifying the most suitable fermentation medium and conditions for EPS production by KM-502.

Carbon sources are indispensable energy and nutrient sources for the cellular synthesis of biopolysaccharides [25]. Previous studies by Yu Wang et al. [26] and Haiyun Yang et al. [27] have demonstrated the beneficial effects of sucrose as a carbon source in the medium for microbial EPS production. Our study corroborated these findings, revealing that sucrose was the optimal carbon source for EPS production by KM-502, consistent with previous reports highlighting sucrose as the most suitable carbon source for biomass and EPS production [28].

In addition to carbon sources, nitrogen sources also play a pivotal role in influencing the yield and biological activity of cellular biopolysaccharides [29]. As primary raw materials for nucleic acid, protein, and enzyme biosynthesis in microbial fermentation, nitrogen sources are vital for microbial growth and development [30]. Common nitrogen sources in microbial culture include peptone, yeast extract, ammonium sulfate, and ammonium nitrate. Our analysis of the effects of these nitrogen sources on EPS production by KM-502 indicated that peptone was the optimal nitrogen source, whereas a 1:1 combination of peptone and ammonium sulfate limited EPS production.

Furthermore, inorganic salts exert certain effects on cellular signal transduction, enzyme catalysis, and transmembrane transport [31]. In our study, we compared and analyzed the effects of NaCl, CaCl_2_, and MgSO_4_ on EPS synthesis by KM-502 and identified CaCl_2_ as the optimal inorganic salt ion. By integrating the single-factor effects of carbon source, nitrogen source, and inorganic salt on EPS production by KM-502, we further optimized the fermentation medium using RSM, resulting in the optimal fermentation medium for EPS production by KM-502: 8% sucrose, 1.99% peptone, and 0.13% CaCl_2_.

Fermentation conditions can influence the activity of enzymes involved in the polysaccharide production metabolism of a strain. Through single-factor experiments and RSM optimization, we determined the optimal fermentation conditions: a medium volume of 74 mL/300 mL, a pH of 6.7, and an inoculum size of 1.99%. When comparing the effects of optimizing fermentation conditions on microbial EPS production, the conclusions of this study differed significantly from the response surface methodology (RSM) optimization results for *Ganoderma lucidum* EPS production reported by Tikhomirova et al. [32]. Specifically, Tikhomirova et al.’s study increased *G. lucidum* EPS yield from 121.1 ± 10.2 mg/L to 229.0 ± 20.3 mg/L through fermentation condition optimization, representing an approximate 89% increase. In contrast, after optimizing the fermentation system in this study, the EPS production of *K. marxianus* KM-502 increased by 22.77-fold compared to the pre-optimization level. Furthermore, compared with other fungi, the EPS yield of KM-502 was higher than that of *Cordyceps sinensis* (CCRC 36421) [33] and *C. sinensis* UM01 [34].

Free radicals are continuously generated and cleared in the body. If free radicals accumulate excessively, free radical-induced oxidative stress can lead to cellular damage, accelerated aging, and various diseases, including cancer, cardiovascular diseases, inflammation, Parkinson’s disease, and Alzheimer’s disease [35,36]. EPS have been confirmed as critical natural free radical scavengers in the dietary antioxidant system, exerting irreplaceable biological functions in protecting against oxidative stress-induced damage [24]. Studies have demonstrated that EPS produced by animals, plants, and microorganisms possess strong antioxidant potential [37]. Among these, EPS produced by microorganisms have garnered widespread attention from researchers due to their diverse biological activities [4].

In this study, we extracted and purified polysaccharides KM1 and KM2 from the fermentation broth of *K. marxianus* KM-502 and conducted in vitro antioxidant analyses. To investigate the antioxidant capacities of polysaccharides KE1 and KE2, we employed tests for hydroxyl radical scavenging activity, DPPH scavenging activity, and Fe^2+^-reducing activity. The results demonstrated that KE1 and KE2 exhibit significant antioxidant activities, with KE2 displaying significantly higher antioxidant activity than KE1. The antioxidant capacities of both polysaccharides were concentration-dependent, similar to those of the antioxidant polysaccharide GCP-WS produced by *Chaetomium globosum* CGMCC 6882 [38].

Hydroxyl radicals are highly reactive and can damage adjacent biomolecules, leading to cancer and various diseases. Therefore, scavenging hydroxyl radicals is crucial for preserving normal cellular homeostasis [39,40]. At a polysaccharide concentration of 10 mg/mL, the hydroxyl radical scavenging activity of *K. marxianus* KM-502 EPS exhibited comparable activity to that of polysaccharides from *Morchella esculenta* [41]. DPPH (1,1-Diphenyl-2-picrylhydrazyl) is a commonly used reagent for in vitro antioxidant assays, exhibiting high stability under ambient conditions and reacting rapidly with free radicals in solution [42]. At a polysaccharide concentration of 8 mg/mL, the DPPH scavenging capacities of KE1 and KE2 were 63.01% and 83.23%, respectively, which were higher than those of PACI-1 (from *Cordyceps sinensis* S1 strain) at 56.46% and FVR-1 (from *Flammulina filiformis*) at 47.38% [43,44]. Furthermore, in terms of Fe^2+^-reducing activity, KE1 and KE2 exhibited higher Fe^2+^-reducing capacities than those of *Sargassum* polysaccharides [45] and *M. esculenta* polysaccharides [41].

Investigating the chemical structure of polysaccharides contributes to elucidating their biological activities, thereby advancing the development of their potential pharmaceutical value [46]. The antioxidant capacity of polysaccharides is subject to multifactorial synergistic regulation by structural and physicochemical properties [43]. Studies indicate that their biological activities, including free radical scavenging capacity, are closely correlated with chemical structural parameters such as monosaccharide composition, molecular weight, glycosidic bond types, and spatial conformation [47]. These structural characteristics collectively determine the unique three-dimensional architecture of polysaccharides, where the type, quantity, and glycosidic linkage patterns of monosaccharides directly influence their distinctive structural features [48]. For example, the backbone of *G. lucidum* polysaccharides (GLPs) primarily consists of β-D-Glcp and partial α-D-Galp. If the side chain contains β-D-Glcp, it exhibits antioxidant activity; if it contains α-L-Fucp, it demonstrates immunomodulatory activity [49]. It has been reported that exopolysaccharides with high rhamnose content exhibit enhanced antioxidant capacity [50]. Notably, the antioxidant capacities of exopolysaccharides KE1 and KE2 in this study cannot be attributed to a single structural factor but rather stem from synergistic interactions among multiple factors. Further investigation is required using techniques such as high-performance liquid chromatography (HPLC), scanning electron microscopy (SEM), atomic force microscopy (AFM), and infrared spectroscopy [51].

## 5. Conclusions

In summary, this study optimized the composition of the liquid fermentation medium for EPS production by KM-502 using single-factor experiments and RSM. The optimal fermentation medium composition was determined to be 8% sucrose, 1.99% peptone, and 0.13% CaCl_2_, with the optimal fermentation conditions being a medium volume of 74 mL/300 mL, a pH of 6.67, an inoculum size of 1.99%, a temperature of 30 °C, a shaking speed of 160 r/min, and a fermentation duration of 4 days. After optimization, the EPS yield of KM-502 reached 5842.42 mg/L, an increase of 22.77-fold compared to the pre-optimization yield. The EPS derived from KM-502, namely KE1 and KE2, exhibited significant in vitro antioxidant activities. This study provides theoretical guidance for the development of natural antioxidant products using *K. marxianus*.

## Figures and Tables

**Figure 1 foods-14-02796-f001:**
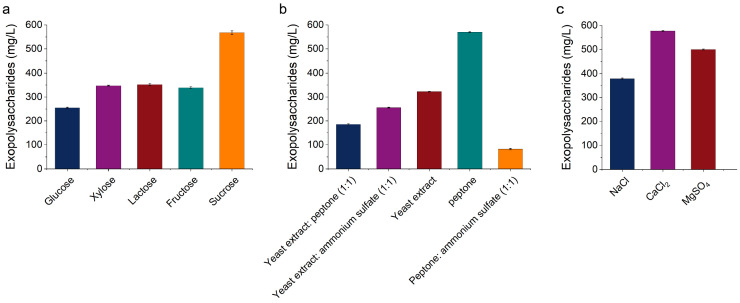
The impact of single-factor screening of liquid fermentation medium components on the extracellular polysaccharide yield of KM-502. (**a**) Carbon source; (**b**) nitrogen source; (**c**) inorganic salt.

**Figure 2 foods-14-02796-f002:**
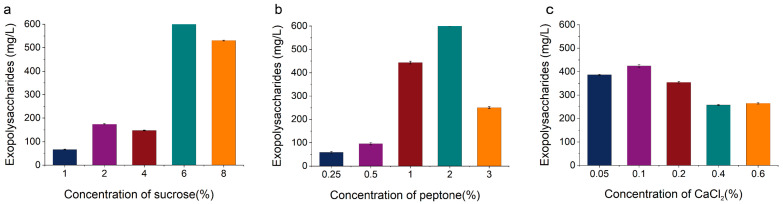
The impact of single-factor concentrations in liquid fermentation medium on the extracellular polysaccharide yield of KM-502. (**a**) Sucrose; (**b**) peptone; (**c**) CaCl_2_.

**Figure 3 foods-14-02796-f003:**
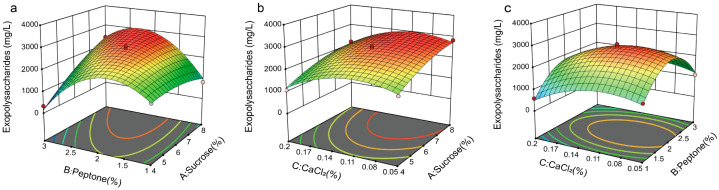
Analysis of the Box–Behnken design for each different component in extracellular polysaccharide production via KM-502 liquid fermentation. The chart’s hue shifts from blue to red, signifying an increase in extraction quality. Among the colored data points, red points represent experimental groups where actual production exceeds the model-predicted value, while pink points indicate groups where actual production is lower than the predicted value. (**a**) Sucrose and peptone; (**b**) sucrose and CaCl_2_; (**c**) peptone and CaCl_2_.

**Figure 4 foods-14-02796-f004:**
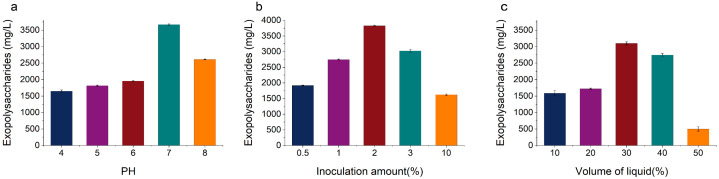
Optimization of fermentation conditions for extracellular polysaccharide production by liquid fermentation of KM-502. (**a**) pH; (**b**) inoculation amount; (**c**) volume of liquid.

**Figure 5 foods-14-02796-f005:**
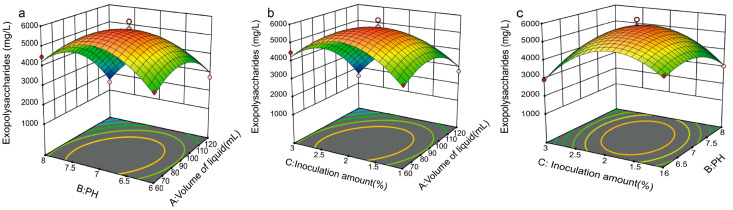
Analysis of the Box–Behnken design for different fermentation conditions in extracellular polysaccharide production via KM-502 liquid fermentation. The chart’s hue shifts from blue to red, signifying an increase in extraction quality. Among the colored data points, red points represent experimental groups where actual production exceeds the model-predicted value, while pink points indicate groups where actual production is lower than the predicted value. (**a**) Volume of liquid and pH; (**b**) volume of liquid and inoculation amount; (**c**) pH and inoculation amount.

**Figure 6 foods-14-02796-f006:**
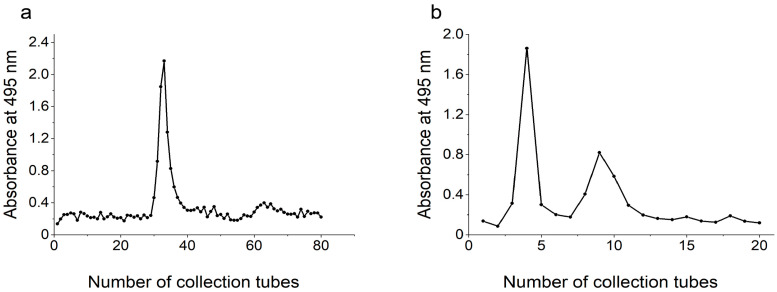
Purification of extracellular polysaccharides from KM-502. (**a**) DEAE-52 cellulose column chromatography; (**b**) Sephadex G-100 column chromatography.

**Figure 7 foods-14-02796-f007:**
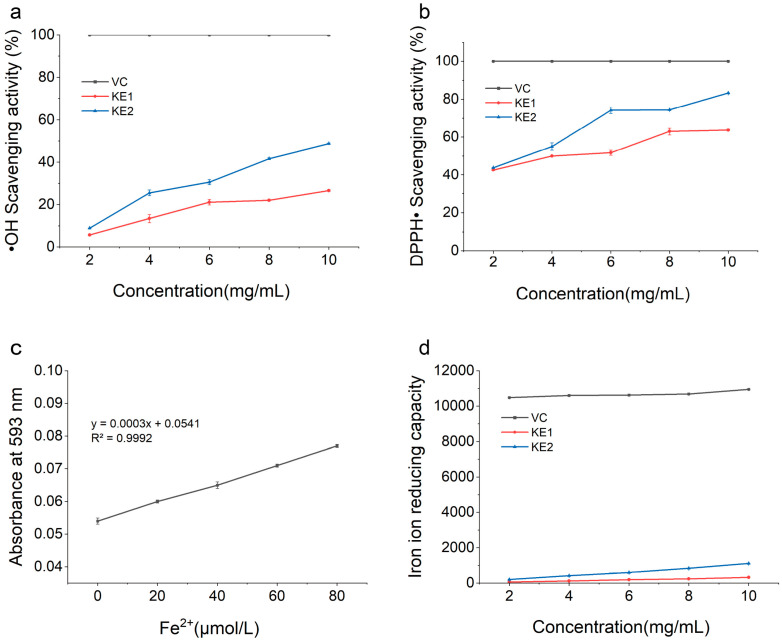
In vitro antioxidant analysis of extracellular polysaccharides KE1 and KE2 of KM-502. (**a**) Analysis of hydroxyl radical scavenging activity of extracellular polysaccharides. (**b**) Analysis of DPPH scavenging activity of extracellular polysaccharides. (**c**) Standard curve of Fe^2+^ reduction ability. (**d**) Analysis of Fe^2+^ reduction activity of extracellular polysaccharides.

**Table 1 foods-14-02796-t001:** Results of Box–Behnken design for optimization of liquid fermentation medium of KM-502.

Std.	Run	Factor 1: Source%	Factor 2: Peptone%	Factor 3: CaCl_2_%	Response 1: Exopolysaccharide Concentration mg/L
1	4	4	1	0.125	1367.74
2	13	8	1	0.125	820.32
3	12	4	3	0.125	328.88
4	7	8	3	0.125	2400.08
5	3	4	2	0.05	1684.31
6	2	8	2	0.05	2833.88
7	11	4	2	0.2	1079.57
8	6	8	2	0.2	2139.89
9	14	6	1	0.05	1267.42
10	17	6	3	0.05	1082.27
11	9	6	1	0.2	592.37
12	10	6	3	0.2	597.687
13	1	6	2	0.125	2741.52
14	15	6	2	0.125	2833.59
15	8	6	2	0.125	2703.4
16	16	6	2	0.125	2597.55
17	5	6	2	0.125	2626.28

**Table 2 foods-14-02796-t002:** Analysis of variance (ANOVA) for the regression model of liquid fermentation medium optimization of KM-502.

Source	Sum of Squares	df	Mean Square	F-Value	*p*-Value	
**Model**	1.297 × 10^7^	9	1.441 × 10^6^	62.39	<0.0001	significant
A-Sucrose	1.743 × 10^6^	1	1.743 × 10^6^	75.43	<0.0001	
B-Peptone	16,296.14	1	16,296.14	0.7054	0.4287	
C-CaCl_2_	7.554 × 10^5^	1	7.554 × 10^5^	32.70	0.0007	
AB	1.714 × 10^6^	1	1.714 × 10^6^	74.21	<0.0001	
AC	1991.09	1	1991.09	0.0862	0.7776	
BC	9069.71	1	9069.71	0.3926	0.5508	
A^2^	1.872 × 10^5^	1	1.872 × 10^5^	8.10	0.0248	
B^2^	6.688 × 10^6^	1	6.688 × 10^6^	289.52	<0.0001	
C^2^	1.298 × 10^6^	1	1.298 × 10^6^	56.18	0.0001	
**Residual**	1.617 × 10^5^	7	23,100.98			
Lack of Fit	1.262 × 10^5^	3	42,065.19	4.74	0.0835	not significant
Pure Error	35,511.30	4	8877.83			
**Cor Total**	1.313 × 10^7^	16				

**Table 3 foods-14-02796-t003:** Results of Box–Behnken design for optimization of liquid fermentation conditions of KM-502.

Std.	Run	Factor 1: Volume of Liquid %	Factor 2: pH	Factor 3: Inoculation Amount %	Response 1: Exopolysaccharide Concentration mg/L
1	1	20	6	2	5180.06
2	9	40	6	2	2483.62
3	8	20	8	2	3739.54
4	7	40	8	2	4014.76
5	10	20	7	1	3741.93
6	13	40	7	1	2745.87
7	16	20	7	3	4474.89
8	6	40	7	3	1482.35
9	17	60	6	1	4124.54
10	3	60	8	1	3041.3
11	12	60	6	3	2926.94
12	15	60	8	3	3296.03
13	14	60	7	2	5980.57
14	2	60	7	2	5233.54
15	11	60	7	2	5545.68
16	5	60	7	2	5585.22
17	4	60	7	2	5586.25

**Table 4 foods-14-02796-t004:** Analysis of variance (ANOVA) for the regression model of fermentation condition optimization of KM-502 strain.

Source	Sum of Squares	df	Mean Square	F-Value	*p*-Value	
**Model**	2.663 × 10^7^	9	2.959 × 10^6^	29.99	<0.0001	significant
A-Volume of liquid	5.136 × 10^6^	1	5.136 × 10^6^	52.04	0.0002	
B-PH	48,598.71	1	48,598.71	0.4924	0.5055	
C-Inoculation amount	2.714 × 10^5^	1	2.714 × 10^5^	2.75	0.1412	
AB	2.208 × 10^6^	1	2.208 × 10^6^	22.37	0.0021	
AC	9.965 × 10^5^	1	9.965 × 10^5^	10.10	0.0155	
BC	5.273 × 10^5^	1	5.273 × 10^5^	5.34	0.0541	
A^2^	4.076 × 10^6^	1	4.076 × 10^6^	41.30	0.0004	
B^2^	2.355 × 10^6^	1	2.355 × 10^6^	23.86	0.0018	
C^2^	9.362 × 10^6^	1	9.362 × 10^6^	94.86	<0.0001	
**Residual**	6.908 × 10^5^	7	98,690.20			
Lack of Fit	4.093 × 10^5^	3	1.364 × 10^5^	1.94	0.2652	not significant
Pure Error	2.815 × 10^5^	4	70,384.90			
**Cor Total**	2.733 × 10^7^	16				

## Data Availability

Data will be made available on request.

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
