# Peer review of "A Study on the Extraction, Fermentation Condition Optimization, and Antioxidant Activity Assessment of Polysaccharides Derived from Kluyveromyces marxianus"

_foods, 2025, doi:10.3390/foods14162796_

Round 1

Reviewer 1 Report

Comments and Suggestions for Authors

The authors optimized the culture medium and fermentation conditions for K. marxianus KM502, extracted and purified two exopolysaccharides from the culture medium, and evaluated their antioxidant activity through in vitro assays. The manuscript is well-organized, and the results are generally consistent with the objectives and content presented. While the optimization process and EPS isolation are particularly relevant and well-executed, the antioxidant activity assays appear to provide only preliminary data. Therefore, further in-depth investigation and validation are recommended in future studies. Regarding the manuscript, just a few comments: In the Materials and Methods section, the procedure for calculating the percentage of inhibition in the in vitro antioxidant assays is not clearly described. Moreover, the rationale for selecting vitamin C as a control is as a control is unclear, especially considering the high concentrations employednot well-justified, particularly given the high concentrations used, which may limit meaningful comparisons. Providing a clearer explanation of this choice would enhance the interpretation of the results. Additionally, a more detailed conceptual framework would improve the clarity of the graphs and contribute to a more comprehensive understanding of the findings.

Author Response

请参阅附件。

Reviewer 2 Report

Comments and Suggestions for Authors

The manuscript addresses an interesting biotechnological topic — the optimization of exopolysaccharide (EPS) production by Kluyveromyces marxianus and the evaluation of their antioxidant activities. The subject is timely and may have practical applications. However, the manuscript suffers from numerous serious shortcomings that, in the reviewer’s opinion, must be addressed before the paper can be considered in Foods:

  1. The Introduction gives the impression that the authors are citing literature randomly, without attempting to provide scientific justification or interpretation.
  2. A clearly stated aim or research hypothesis is missing.
  3. Regarding strain KM-502 — the authors should specify its origin, whether it has been deposited in a public microbial culture collection, and provide its accession number and/or GenBank sequence. Is the strain available to other researchers?
  4. The “Discussion” section (lines 447–512) is essentially a restatement of the results, with no reference to underlying biological mechanisms or critical analysis.
  5. There is a lack of deep interpretation or comparison of the findings with other studies in the field.
  6. No structural characterization or molecular weight data for the EPS are provided, despite their relevance to antioxidant activity.
  7. Why are all words capitalized in the legend of Figure 3? This is not consistent with scientific formatting.

Below are additional, specific technical comments:

– Lines 14–15: The sentence “their abilities to hydroxyl radical scavenging activity, DPPH scavenging activity, and Fe2+ reducing activity” is grammatically incorrect. Suggested revision: “their hydroxyl radical scavenging, DPPH scavenging, and Fe2+ reducing activities were evaluated.”

– Line 17: The phrase “Through single-factor experiments and response surface methodology…” is vague. A clearer formulation would be: “The fermentation conditions were optimized using single-factor experiments followed by response surface methodology (RSM).”

– Line 22: The reported value “74.302 mL” is excessively precise for biological experiments. The authors are advised to round to “74 mL”.

– Line 70: The expression “provide a theoretical reference for the promotion…” is vague and unscientific. A more appropriate phrasing would be: “provide experimental evidence supporting potential applications…”

– Lines 83–86: The cultivation conditions are not clearly specified (e.g., duration, temperature, shaking speed). Please revise to include information such as: “incubated at 30°C with shaking at 160 rpm for 24 h.”

– Line 295: The statement “this model can be used to analyze…” is a repetition of earlier content and adds little to the interpretation of results.

– Lines 300–304: The statement “the interaction effect of BC was greater than that of AC…” is a dry statistical observation. A biological explanation should be added — why are these interactions relevant?

– Lines 331–335: In the interpretation of pH effects, the authors state “It is speculated that…”, but no literature is cited to support the proposed link between pH, membrane permeability, and enzyme activity. Such references should be included.

– Lines 383–384: The values “1.987% inoculum” and “pH 6.685” are artificially precise. Given the inherent biological variability, these values should be rounded.

– Lines 417–426: The manuscript does not include statistical data to confirm the significance of differences in antioxidant activities between KE1 and KE2.

– Lines 447–492: The Discussion section largely reiterates the results and lacks in-depth analysis of biological mechanisms or comparisons with EPS-producing yeast strains described in the literature.
